# Ecomindsponge: A Novel Perspective on Human Psychology and Behavior in the Ecosystem

**Minh-Hoang Nguyen** [1,2], **Tam-Tri Le** [1,2,*] and **Quan-Hoang Vuong** [1]

1    Centre for Interdisciplinary Social Research, Phenikaa University, Hanoi 100803, Vietnam
2    A.I. for Social Data Lab (AISDL), Vuong & Associates, Hanoi 100000, Vietnam
*    Correspondence: tri.letam@phenikaa-uni.edu.vn

**Abstract:** Modern society faces major environmental problems, but there are many difficulties in studying the nature–human relationship from an integral psychosocial perspective. We propose the ecomind sponge conceptual framework, based on the mindsponge theory of information processing. We present a systematic method to examine the nature–human relationship with conceptual frameworks of system boundaries, selective exchange, and adaptive optimization. The theoretical mechanisms were constructed based on principles and new evidence in natural sciences. The core mechanism of ecomindsponge is the subjective sphere of influence, which is the limited mental representation of information received from and processed based on the objective sphere of influence–actual interactions in reality. The subjective sphere is the sum of two sub-spheres: influencing (proactive) and being influenced (reactive). Maladaptation in thinking and behavior of the mind as an information collection-cum-processor results from the deviation of the subjective sphere from reality, which includes two main types: "stupidity" and "delusion". Using Bayesian Mindsponge Framework (BMF) analytics on a dataset of 535 urban residents, we provide consistent statistical evidence on the proposed properties of subjective spheres. The dynamic framework of ecomindsponge can be used flexibly and practically for environmental research as well as other psychosocial fields.

**Keywords:** environmental research; nature–human relationship; information processing; subjective sphere of influence; Bayesian Mindsponge Framework

## 1. Introduction

### 1.1. A Demand for a Conceptual Framework Connecting Humans and Nature

The human brain is a highly evolved biological organ with an exceptional capacity for information processing. With this advantage over other species on Earth, humans are able to have a significant impact on both the environment and each other. We have established advanced civilizations with flourishing cultures. Humanity is familiar with its capacity to create complex social structures, including artificial laws and complex norms. We also frequently think that creativity is what distinguishes us from other creatures. In fact, many people may believe that human society holds principles completely different from those of the natural world. However, where do we stand regarding the Earth and the rest of the universe? How much influence do we have over others and ourselves?

The ecosphere is being severely harmed by humans, who are currently contributing to the ongoing Sixth Mass Extinction [1,2]. These effects are detrimental to human health as well as the survival of our species and other living things. A species' information processing system may be considered seriously flawed when it exhibits chronic self-harm behaviors and destroys its own habitat. Despite having more knowledge than any other species on Earth and a central nervous system superior to them, why are we acting in this way?

Scientists and policymakers are aware of the various impacts of nature on humans and vice versa. Social sciences are needed to address environmental issues, and understanding human behavior is necessary to prevent climate change and conserve biodiversity [3,4].

The scope of social sciences is expanding to fulfill the growing demand for knowledge on how humans and nature interact. Major theories, such as Cognitive Load Theory [5], Theory of Planned Behavior [6], Adaptation-level Theory [7], Maslow's theory of human motivation [8], and Bounded Rationality Theory [9], are making significant contributions to our understanding of how people behave in real-world settings, but they frequently concentrate on just one component of how people process information and they may lack the flexibility needed to be applied to multiplex or extreme cases of ideation and behavior. For example, studying the psychology of climate change deniers and other related conspiracy theories, hypocrisy, active indirect self-harm, cognitive dissonance tolerance, etc., requires a deeper understanding of the underlying psychological mechanisms with multi-layered interactions between involved factors [10].

Currently, a lot of environmental policy failures reflect the yet-to-be-known boundaries between thinking (planning, frameworks, etc.) and reality (public behavior, physical impacts, etc.) [3]. A more dynamic and adaptive framework of information processing can take advantages of the understanding from existing theories and expand it to better fit the ever-changing self-regulating systems of the natural, social, and mental worlds.

*1.2. Human Thinking and Reality*

What is reality? If a baby had only been exposed to virtual reality from the moment of birth, would there be a realization about the environment's properties of being real or fake? The famous movie series The Matrix, an artistic presentation of the intriguing yet debatable simulation hypothesis [11], may remind us to contemplate the "realness" of reality which is often taken for granted. Characteristics of observed objects or phenomena are "known" by the self in the form of qualia [12]. In a sense, the awareness of perceptions is always indirect, where perceptions are products of information processes that are presented to the self. The self (or ego) is a mental construct (another set of processes running simultaneously) and the center node for all self-interested filtering processes. Within the scope of neural signaling, the work of the mind is objective brain activities. Within the scope of conscious awareness (as results of base activity), thinking is subjective. These interpretations are also in alignment with the relativistic view on consciousness proposed by Lahav and Neemeh [13]. After all, the human brain, despite being a very complex system, is a product of more fundamental physical interactions and thus follows the laws that govern such interactions. This way of reasoning is based on the anthropic principle of nature [14].

The nature–human relationship is not straightforward, and many interacting pathways are largely unknown. The biophilia hypothesis was proposed by Wilson [15], suggesting that humans have an innate tendency to connect with nature. Exposure to nature was largely found to be associated with positive emotions and psychological, physiological, and cognitive benefits [16,17]. In terms of neurological pathways, the perceived benefits of exposure to nature can be attributed to its impacts on the brain. A walk in nature can decrease self-reported rumination and neural activity in the prefrontal cortex that may be linked to depression [18]. More recently, further studies have found that this phenomenon may be due to how nature exposure supports healthy amygdala structures [19] and decreases amygdala activation, which is often linked to stress responses [20].

Sometimes, the attraction to nature is due to hardwired biological needs. For example, a study found that humans' preference for glossiness likely stem from our physiological desire for water [21]. Other times, the pathways may even involve other species co-existing with the human body. For example, the health benefits of nature exposure can be partially attributed to the complex interactions between the human body and microorganisms that support healthy immune system development [22]. Interestingly, it should also be noted that gut bacteria can affect human cognitive processes through pathways of neurotransmitters [23,24].

How people perceive things is dependent on the objective properties of those things, the subjective mindsets of the observer, and how the information is transmitted and processed. In the presentation throughout this study, the term perception is used in its broadest sense (what one thinks about something), including both basic sensory perceptions and more

advanced cognitive processes. The mind changes as it functions and interacts with new information from the environment. Such changes are adaptations enabled by the natural property of neuroplasticity [25]. For example, perceptions of objective measurements in the external environment can change depending on the sense of control and expected outcomes of actions [26]. To effectively study the highly dynamic information processes of the human mind, we need a compatible method derived from the same natural principles that create the brain while also being highly flexible to fit diverse psychosocial contexts.

*1.3. A New Information Processing-Based Approach*

The mindsponge theory [27] offers a novel approach and an effective framework for exploring the nature–human relationship down to the deepest levels of information exchange and system interactions. The mindsponge mechanism was originally conceptualized by [28] to explain how the mind filters new values, accepting those in alignment with existing core values and rejecting those in contrast. As an information collection-cum-processor, the human mind filters inputs using subjective cost–benefit judgments following objective natural biochemical and physical principles, then generate outputs as thoughts and other neural signals responsible for corresponding behaviors. These information processes happen non-stop and change the mindset and the filters based on newly integrated values, thus updating the whole system for adaptation [10]. This reflects the constant self-regulation and self-optimization of every system in the biosphere, which shares the same fundamental functioning principles of the human mind.

The mindsponge information processing mechanisms of the mind serve as effective tools to investigate the nature–human relationship deeply. However, the mindsponge theory, in its general form, lacks conceptual boundaries that help increases its applicability when examining human thinking in relation to nature. For example, these boundaries include but are not limited to those between objective reality and mental representations, between ideas and behaviors, and between the natural and the artificial. Moreover, based on the views of physics and using mathematical reasoning, Lahav and Neemeh [13] suggested that consciousness is relativistic, meaning that it is neither non-reductive nor illusional but rather dependent on the observer. Thus, to determine the position and scope of observation is to find the boundaries of concepts and phenomena.

Additionally, the human brain and human society are much more complex than other species on the planet. It is helpful to draw analogies from other organisms, but the human mind's activity needs to be considered in a higher-level framework. There are no other known types of advanced processing systems on an equal level to the human mind. Thus, developing a conceptual approach to studying human subjectivity is needed since direct comparison with another system is largely impossible.

Therefore, we propose the ecomindsponge conceptual framework to focus on examining the human information processing system in relation to its external environment–the Earth's ecosphere. The framework goes deep into the fundamental subject–object relationship and the mechanisms of dynamic adaptation in a system. Psychosocial patterns are linked to their corresponding foundational principles in physics and biochemistry. Nature–human interactions are examined in terms of information exchange and system optimization.

## 2. Ecomindsponge Conceptual Framework

In this section, we provide the general definition and explanation for the ecomindsponge conceptual framework. Generally speaking, ecomindsponge is an information processing conceptual framework of how humans make sense of nature through their subjective perceptions and process feedback from interactions with the environment. There are two fundamental concepts in the conceptual framework:

1.   The objective sphere of influence;
2.   The subjective sphere of influence.

The objective sphere of influence demonstrates the physical interactions between a human and other component of reality, including other humans. The conceptualization

is based on the Weak Anthropic Principle [14]. It is important to set the boundary for the concept of "information" used throughout our arguments. We use the metaphysical perspective positing that everything in the universe is information [29–31]. A human is a component of the Earth's ecosphere, a very large-scale natural information processing system. Other information processors and the information in our surroundings exert influences on us, but we also affect them through reactions. From a physical point of view, in the continuous spacetime of reality, it can be considered that all actions are reactions [32]. The interactions between humans and their surrounding environment in the physical world can be deemed objective interactions, bound by the parameters, constants, and natural laws; thus, they have certain limits. The set of all interactions under said boundaries is the objective sphere of influence.

Planet Earth is a colossal information processing system. Humanity as a species and human society are part of this planet-level system. Humans interact with the natural environment, including other humans, on a physical level. At the same time, the mental representation of each person ("self" as a mental construct) also interacts with other mental representations of various objects and phenomena in reality. In fact, it is suggested that in the evolution of human cognition, mental representations of hidden variables allow causal cognition to drive advanced learning and technological advancement [33]. The widely debated philosophical issue of subject–object duality should be viewed as a more dynamic relationship.

Besides the notion of consciousness' relativity [13] based on relativistic physics [34], the information-based approach to this dualistic relationship should be accompanied by a reductionistic view [35,36], especially regarding the biological aspect of beings [37]. From the standpoint of neuropsychology, perceived awareness can be considered the result of a schematic model of the process of attention constructed by the brain [38]. Fundamentally, the structures of biological beings are all built from atoms. More complex molecules and their interactions cause more complex structures and functions. The human body, including the brain, is what enables mental simulations. We currently do not understand or are not even are aware of many mechanisms that created the sense of subjectivity. However, everything is built on the laws of reality, whether we know them or not. In a sense, it is important to acknowledge the countless objective interactions between the human mind and various other systems (other "minds") in the natural environment.

To have a better understanding going into examining the nature–human relationship, it is helpful to briefly look at the evolutionary process leading to the physical structures that create our "subjectivity". This can help provide a clearer picture of the connections that support information exchange as well as the processing system's principles and constraints.

A highly advanced nervous system, such as in humans, has its ancestry as basic mechanisms found in simple eukaryotes. Specifically, one of the simplest examples is the $Ca^{2+}$ channels in choanoflagellates [39]. These unicellular eukaryotes have the Munc18/syntaxin1 neural protein pair which controls neurotransmitter release similar to vertebrates' neurons [40]. Signal transduction mechanisms for extracellular communication are what drive the evolution of more complex neural systems, from neural networks in Hydra and simple central nervous systems in planarians to highly expanded cerebellar and cerebral cortex in humans that support advanced cognition [41–43].

The human brain's functions are based on the activities of neurons and their synapses, which in turn follow the biochemical principles of molecular interactions [44]. Our social perceptions and behaviors are the results of information processes happening in various regions of the cerebral cortex [45]. It is worth noting that many brain regions, as well as other parts of the body, contribute to our thinking. For example, cost-benefit decision-making involves how the ventral striatum responds to the expectation of pain or reward [46,47]. Another typical example is how gut bacteria can affect a person's state of mental health, such as depression [23,48] due to their ability to produce neurotransmitters including dopamine, norepinephrine, serotonin, etc. that affect the human brain [24].

Activities and adaptation tendencies of biological systems are dependent on usable resources, especially in terms of energy expenditure [49]. For eukaryotes, such energy sources

come from adenosine triphosphate (ATP) molecules which are produced mainly through oxidative phosphorylation in mitochondria or photophosphorylation in chloroplasts. The issue of energy constraint is particularly important when it comes to the human brain, as it accounts for only 2% of the body mass but consumes about 20% of total oxygen and calorie [50]. During the intensive development period in children, the brain may take up to half of the body's energy [51]. A baby's brain has about $10^{14}$ synapses, which are trimmed down to half that number through maturation, where unused synapses are eliminated in favor of strengthening well-used circuits [25].

Furthermore, the neural wiring of the human brain's neocortex evolved in a way that allows it to perform complex functions with minimal energy expenditure [52]. Another example of such physiological influences on thinking is that prioritization of energy input for survival purposes makes the ventral striatum, amygdala, anterior insula, and medial and lateral orbitofrontal cortex biased toward high-calorie food items [53]. Similar to the biological evolution of enzymes for increasing energy efficiency [54], advanced cognition in humans also employs energy-saving mechanisms in information processing such as the use of trust [27,55].

"Minds"–systems of living beings, including human minds, are defined as information collection-cum-processors [27]. However, the lack of system boundaries hinders the more in-depth investigation into the mind's interactions with its surrounding environment, so the subjective sphere of influence is proposed to overcome this limitation. The subjective sphere of influence is a set of all interactions among information existing within the mind (or the subjective world) under mental boundaries. The subjective sphere of influence reflects the objective interactions that can be perceived and exist as corresponding representations within the mental realm.

Information received from the external environment or generated internally is stored as engrams–cognitive information imprinted in a physical substance–through biochemical changes in neurons. In engram neurons, memory consolidation happens due to DNA (deoxyribonucleic acid) methylation triggered by signals that leads to stable structural changes [56,57]. The hippocampus and the amygdala are believed to be majorly responsible for creating cognitive maps of stored information [58,59]. Information entering the brain is suggested to be stored temporarily in the hippocampus before being consolidated in the neocortex for long-term storage [60,61]. As signals are transmitted through synapses, their connection strength increases in the process of long-term potentiation [62]. These dynamic information storage mechanisms are the basis for neuroplasticity which enables flexible adaptations for the processing system.

Here, it is important to note that the brain is not the only structure capable of dynamic information storage, as memory outside the brain is necessary and common in the biosphere [63,64]. On a broader meaning of memory, the concept becomes apparent when we look at various essential components of life, such as the immune system, genetic material, etc. Even individual cells are found to have dynamic mechanisms of molecular working memory [65]. Overall, natural information storage and transmission mechanisms in a biological system (specifically the human mind) allow it to absorb information from the objective sphere into its subjective sphere.

In the next section, we thoroughly explain the construct and function of the subjective sphere of influence and its relationship with the objective sphere.

### 2.1. Subjective Sphere of Influence

According to the mindsponge theory [27,28], the subjective (mental) world is constructed from a set of information within the mind. Information in the mind interacts with each other, forming webs of associations that generate the subjective sphere of influence. The sphere creates the attached meaning for mental representations of objective things and events. The meaning of something to an individual is due to one's web of associations based on the whole history of one's life experiences [25].

Three fundamental components construct the subjective sphere of influence:

1. Information;
2. Connection between information;
3. The intensity of the connection.

Two prerequisite conditions are required to form a singular instance of the subjective sphere of influence. The first condition is the existence of at least two distinct pieces of information within the mind. The second condition is the existence of at least a connection between those two pieces of information. There are three main types of connection between two pieces of information (presumably, A and B):

- Perceived impact of A on B (A→B);
- Perceived impact of B on A (B→A);
- Perceived mutual impacts between A and B (A↔B).

Each connection also has intensity, reflecting the perceived likelihood of the connection's occurrence.

To further explain how the subjective sphere of influence can help a person navigate in the ecosphere, we assume that there exists information representing the 'self' within the mind (self as a mental construct). This set of information is special as it reflects the existence of the physiological body and mental construct of the individual in the objective world into the subjective world. Such a construct is helpful for the brain in controlling attention and generating a sense of self-awareness [38]. It is noteworthy that the information representing the self is not fixed, but it varies depending on each person's mindset and updating mechanism. Moreover, as the information representing the "self" is only a projection of the individual's physiological body and mental construct, it is incomplete information and cannot fully reflect the whole body or mind. It is also worth noting that the "self" serves as a central node in filtering processes of self-interested values.

Since an individual's mind (as an information-processing system) is bounded by the natural patterns of the biological system, its fundamental purpose is to prolong the system's existence in one way or another, including survival, growth, and reproduction (in both natural and social aspects) [66,67]. Therefore, information processes within the mind (consciously or subconsciously) prioritize values and connections that favor the existence of the self, which underlies the psychological mechanism of self-affirmation [68].

Because of this self-prioritization mechanism, thinking processes and behaviors are mainly driven by the subjective spheres of influence in relation to the self. Two types of such spheres are:

1. Perceived sphere of being influenced, representing the perceived impact of other information on the self;
2. Perceived sphere of influencing, representing the perceived impact of the self on other information.

These two types can exist concurrently and overlap, creating four different basic scenarios of perceptions.

1. **Scenario A** (cyan bubbles): The information perceived to be mutually influential with the self;
2. **Scenario B** (light red bubbles): The information perceived to be influenced by the self;
3. **Scenario C** (purple bubbles): The information perceived to influence the self;
4. **Scenario D** (purple bubbles): The information perceived to have no interaction with the self.

It should be noted that the influence can be perceived as negative or positive depending on the existing information within the mindset and information observed from the surrounding environment, resulting in even more complex scenarios.

Figure 1 displays the four scenarios of perceptions in the infosphere, with the Earth system being the largest information collection, represented by the green area. Within the Earth is the social environment, where the interactions among humans happen.

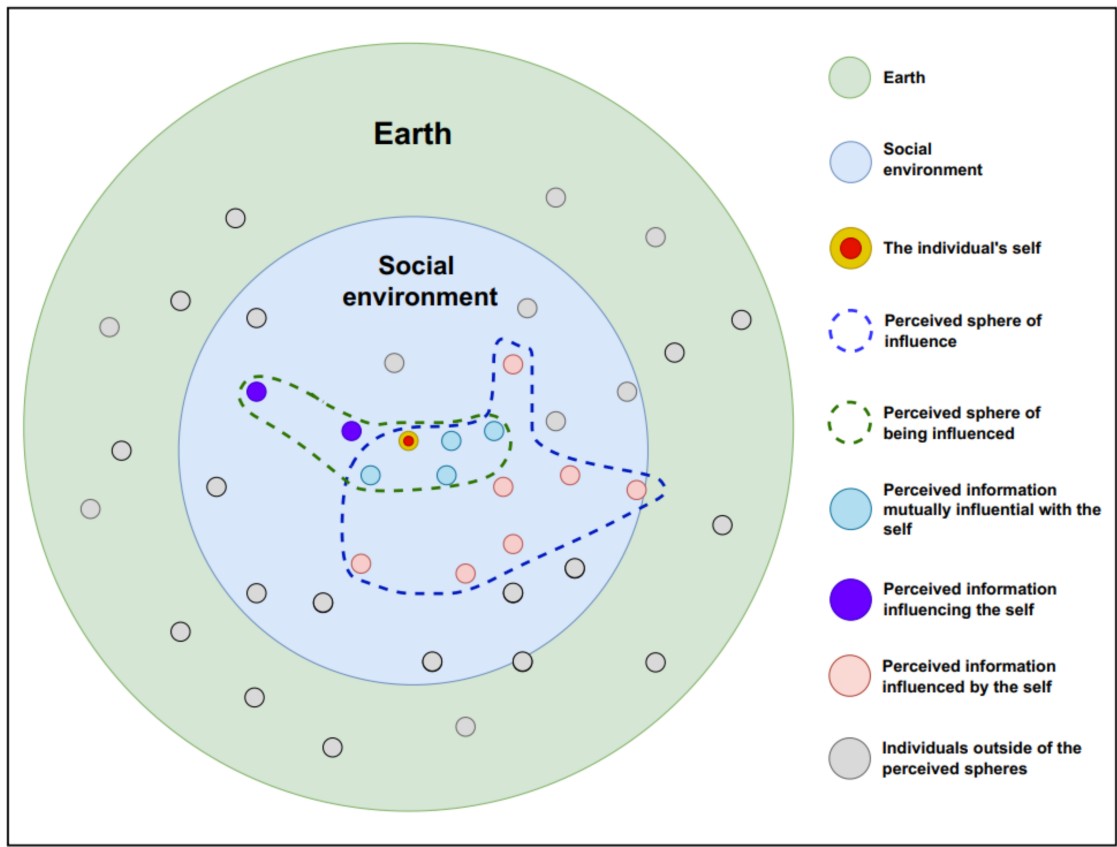

**Figure 1.** Scenarios of perceptions in the infosphere.

The subjective spheres of influence are not only the outcomes but also the inputs of the mental process, supporting the individual in navigating within the infosphere. On the one hand, the shapes of the spheres are dependent on the mind's prior information filtering to maximize the perceived benefit and minimize the perceived cost for the self. On the other hand, they also attribute to future filtering processes of related information. For example, a pleasurable encounter increases favor toward similar interactions in the future and vice versa. Following this way of reasoning, information in **scenario D** holds zero self-interested value and cannot be integrated because it is ejected soon from the mind upon reception for energy-saving purposes.

*2.2. Range of Perception*

The objective sphere includes everything the mind knows, is aware that it does not know, and is not aware of at all. As the subjective sphere expands, more connections are made; thus, the mind becomes aware of more information it did not know. However, a system's range of perception does not expand beyond its physical limitations.

To maintain structural and functional integrity, a system exchanges information with the external environment through selective channels. At cellular levels, the membrane serves not only as a physical boundary and barrier but also biochemical gateways. Transmembrane transport can be highly specific and precise, using the complex mechanism of both passive and active carriage of ions and molecules [69,70]. Similar to cell-level selective permeability, sensory perception absorbs and responds to certain information (not all that is available) in the external environment. In very simple creatures such as bacteria, chemotaxis is used for navigation [71]. Research has also shown that the plant *Arabidopsis thaliana* can detect distinct vibrations caused by caterpillar feeding to activate chemical defense mechanisms [72]. The evolution of information reception channels, such as sensory perceptions, is driven by the pressure of adaptation. For example, trichromatic vision in primates was naturally selected due to its advantages in detecting food over dichromatic

vision, especially regarding fruits [73–75]. In brief, the specific structures and functions of information reception channels in a biological system are optimized for benefits in terms of the system's survival.

In humans, visual perception is a major channel of information input for the brain [76]. Different types of photoreceptors in the eye structure are responsible for receiving information from interactions with photons [77]. Here, a typical example of a natural limitation on information accessibility is that on the electromagnetic spectrum with wavelengths ranging from picometers to megameters, a human can only see the radiation of wavelengths approximately between 400 and 700 nanometers (the range of visible light). Our visual representation of the world is only a small fraction of all the objectively available electromagnetic radiation information. Other biological systems can have different natural ranges of visual perception. For example, many snakes, particularly *Crotalinae* vipers, can detect infrared radiation for hunting prey [78,79]. Compared with the human retina having three cone photoreceptors that produce trichromatic vision, mantis shrimps have twelve photoreceptors responsible for color perception [80].

While humans can use machines and mental simulation to overcome our natural sensory limitations, there are some hard boundaries when it comes to information processing and interpretation. As a casual example, people often jokingly ask if one can imagine a completely new color. Overall, in the process of information reception, the perception of real objects or events is incomplete information, which causes a natural deviation between the subjective sphere and objective sphere. We can think about the metaphor of Plato's Cave.

When examining the content, boundaries, and interactions of systems in the infosphere, we can use the mathematical reasoning of set theory [81]. The objective sphere (base reality) as a collection of information is the set *Environment*. Since human is part of nature, the subjective sphere is the set *Mind*, a subset of the set *Environment*.

$$Mind \subset Environment$$

Assume the existence of an information particle – a bounded carrier (frame) of information [82]. Assuming there is an information particle $A$, If the information particle $A$ does not exist in the set *Environment*, it has zero probability of appearing in the set *Mind*.

$$\forall A(A \notin Environment \implies A \notin Mind)$$

This is the aspect of information availability. Similarly, the same logic can be applied to receivable information (accessibility) and accepted information (filtering and integration).

Besides selective physiological mechanisms, the human mind also employs more advanced methods of information filtering (that often also involve complex simulation or evaluation). Some types of filtering are processed automatically, such as how we are not aware of the natural blind spot in our vision, where the optic nerve passes through the optic disk [83]. Active filtering, such as trusting/distrusting, requires cognitive functions of attaching priorly filtered values to similar information. Trust can be an energy-saving mechanism of information reception and integration that speeds up the filtering process by prioritizing acceptance or rejection [55,84]. Memory storage is energy-intensive, and thus the mind might be biased toward keeping more beneficial information according to subjective evaluation [27]. Such advanced selectivity for system optimization is also shown through the active neurological mechanisms of forgetting [85], as opposed to the former notion that forgetting (ejecting information) is a "fault" of brain function.

### 2.3. Feedback-Induced Updating Mechanism

Due to the way information feedback is processed for updating the mind, there exist differences in the tendencies of responses between the sphere of influence and the sphere of being influenced (also see the subsection "subjective sphere optimization" below for more details on the optimization process).

- The sphere of being influenced shows the passive reaction of humans to nature. The reaction does not have clear long-term strategies and is more dependent on immediate contexts. Passive mindsets tend to have higher degrees of variance in intentions.
- The sphere of influence shows the active action of humans toward nature. Such action has relatively clearer strategies and driving core values (desires). Proactive mindsets tend to have stronger and more consistent intentions.

Thus, patterns of behaviors driven by the sphere of influence are clearer than those driven by the sphere of being influenced. Note that with time and energy constraints, the mind always tries to optimize thoughts and behaviors based on available information and estimated consequences. Fundamentally, these differences in tendencies are shaped by energy expenditure limitations. A simplified interpretation is presented in Table 1. A complete perception with a balanced knowledge of both ways of nature–human interactions is the end goal for public communication endeavors.

**Table 1.** Scenarios of perceived influence.

| | | Sphere of Influence | |
|---|---|---|---|
| | | **Not Exist** | **Exist** |
| Sphere of being influenced | Not exist | Out of perception range | Proactive mindset |
| | Exist | Passive mindset | Complete perception |

The connections within the subjective spheres of influence are mental constructs, so they are not necessarily consistent with their corresponding connections in the objective world. The objective sphere of influence is the baseline in the comparison, including natural parameters, rules, and laws of reality. The subjective sphere, to certain degrees, deviates from the objective sphere due to the variances in the mental processes of the mind (e.g., absorption and simulation processes). The degree of deviation can show how much the subjective sphere fits the objective sphere (see Figure 2). Generally, there are two types of deviations:

1.  Stupidity: the state in which the individual does not understand sufficiently how the ecosphere operates around him/her.
2.  Delusion: the state in which the individual obtains wrong perceptions about the ecosphere operating around him/her.

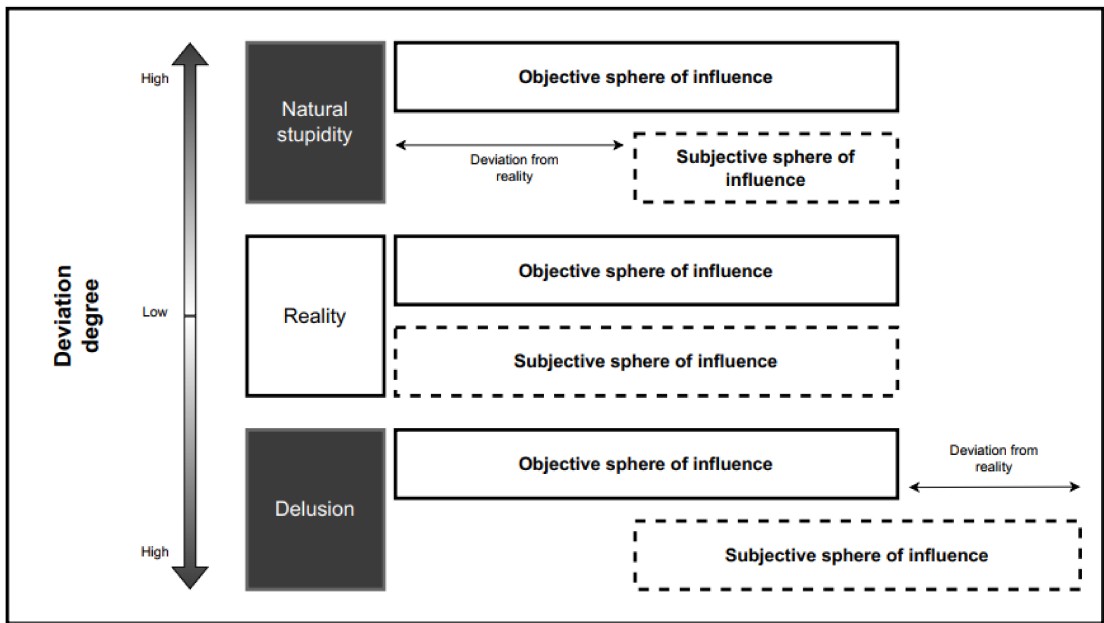

**Figure 2.** The deviation from the reality of humans' perceptions.

In order to construct the subjective spheres and update/adjust them to better fit/reflect the objective sphere, the mind needs to receive and filter new information from reality. Such observation and feedback provide necessary inputs as references for the filtering system to work effectively. These processes require information transmission. Overall, there are two main types of pathways:

- Direct absorption (sensory perceptions) and feedback from behaviors;
- Indirect absorption through other information transmitters.

To demonstrate information absorption from the external environment, we can look at actual situations in contemporary nature-human interactions. People can learn new knowledge or update their existing knowledge by directly experiencing nature. For example, they can visit a national park or tend to their garden and absorb information about nature's properties (e.g., colors, sounds, smells, feelings, etc.) and attach meanings of the interactions (e.g., relaxation, beauty, liveliness, physical tiredness, dangerous species, etc.). Alternatively, they can also learn and reflect using other media of information such as books, videos, verbal conversations with other humans, etc. Urban people in modern society mostly rely on the latter pathway. It is also the dominant choice for conventional environmental education and awareness campaigns. Nonetheless, both types of transmission function based on the trust mechanism. Information is only absorbed and evaluated favorably if it is accompanied by positive trust (as opposed to distrust) [55,84]. In a sense, it is the degree of how "real" the information is deemed (subjectively perceived to be close to one's reality).

*2.4. Subjective Sphere Optimization*

The "desire" to fit the subjective sphere to the objective sphere represents the natural tendency of adaptation to living environments in biological beings–the essential driver of evolution [66]. The representation of the subjective sphere is more like a coil running along and around a central line being the objective sphere. From a two-dimensional view, it looks like fluctuations up and down; but conceptually, the two lines do not cross each other. For easier interpretation, we used simple lines, as presented in Figure 3, to show the dynamic difference between the spheres. Additionally, while the objective sphere line is represented as a straight line as it is the base for comparison with the corresponding subjective sphere, it is worth noting that in reality, objective values also constantly change along the chronological axis.

Figure 3A shows a simplified single optimization process of the subjective sphere, where the subjective value approaches the objective value as time progresses. Similar to the mathematical concept of limit, the subjective line never reaches the objective line. This is because, fundamentally, the subjective value is a result of an interaction ("observation" or information reception) and thus cannot equal the corresponding objective value, which is a component of that interaction. In other words, the value of "changing" in an interaction cannot equal zero. For advanced systems such as the human mind, mental representations (*qualia*) are even more apparently different in nature from the objective things they represent [12]. Due to natural limitations in input channels (e.g., sensory perceptions) as well as processing capability (e.g., deriving the right patterns in observed phenomena), there is always some degree of deviation between the subjective line and the objective line. We call this type of deviation natural stupidity (or simply stupidity). The aim of the optimization process is to reduce this deviation based on feedback from further interactions with reality.

One of the typical examples of such adaptations is how children learn to walk as they carry out thousands of iterations (steps) every day, including many failed attempts. In fact, 12- to 19-month-olds average more than 2300 steps with about 17 falls per hour and make adjustments along the experience to become more effective and efficient at walking [86]. In adults, more complex motor skills also require more frequent social feedback during learning [87].

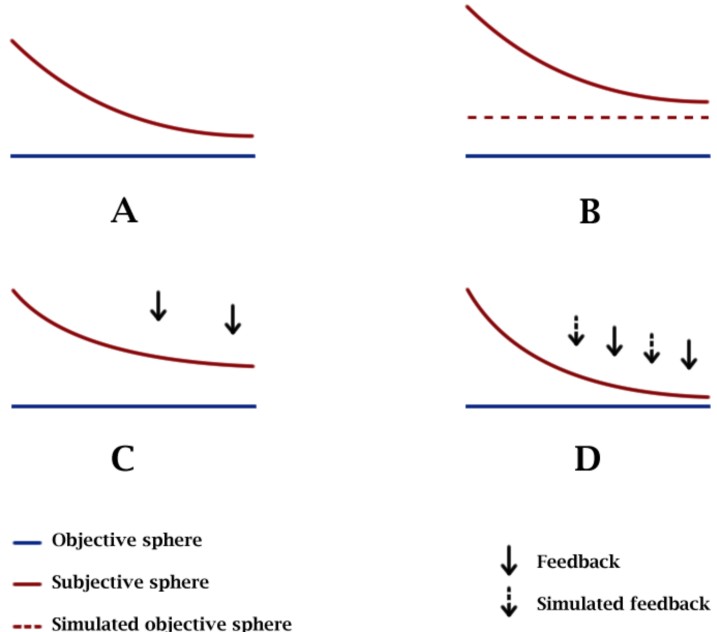

**Figure 3.** Simplified models of subjective sphere optimization processes. (**A**) general process; (**B**) deviated simulated reality; (**C**) reactive approach; (**D**) proactive approach.

Active optimization in advanced systems capable of simulation, however, has a lot of complex information processes that cannot be compared directly with their objective correspondents. A typical example is expectations about future events (e.g., preparing for an interview that is yet to happen). In such cases, the optimization processes use simulated objective values (see Figure 3B). Note that the simulated objective sphere itself is optimized by the mind to fit reality as much as possible in order to create accurate subsequent optimization processes. The simulated line cannot completely fit the objective line due to the same reasons mentioned above. We call the deviation between these lines delusion, which can be the result of natural limitations or malfunctions in the information reception or filtering system (e.g., sensory perception disorders, schizophrenia, use of hallucinogens, blind radical beliefs, etc.). Psychosis affects the whole updating process of adaptation, where experience and learning are deranged and further distort subsequent processing [88].

Optimization occurs by connecting and comparing prior beliefs to new evidence in the form of feedback for adjusting and updating. A system can process feedback from both natural encounters (see Figure 3C) and simulation (see Figure 3D). The reactive approach is heavily dependent on information availability in the immediate surrounding environment. This may result in absent or incomplete feedback (e.g., trying to observe wild boars in the ocean). Other times, learning by direct trial and error may put the system at risk of being harmed (e.g., approaching a wild boar to see how it would react).

Advanced systems have better processing capabilities, allowing simulation to aid in such processes where direct feedback is unavailable. For example, the young human brain learns to "fill in the blank" using mental representations regarding object permanence understanding in infants [89,90]. By using information stored in memory, the human brain can generate mental representations of expected future events, including the projected self (as a mental construct) in those simulated interactions [91,92]. Mental simulation is a helpful function of the human mind for problem-solving tasks [93]. Noteworthily, the brain uses the same neural machinery of remembering the past to simulate the future [94], which also helps explain why memory and imagination are both easily affected by constructive adaptive processes [95].

The proactive approach in optimization, thus, is more efficient at fitting the subjective sphere compared with the reactive approach. In humans, behaviors carried out under self-

control are driven by belief-based intentions [6,10]. Planning involves a considerable degree of future simulation and thus likely makes the product of the information process (thoughts or behaviors) more focused (converged). For example, the responses of people who see a baseball being thrown at them may highly vary (fluctuating around the objectively optimal option in each context), such as dodging, bracing for being hit by the ball, catching the ball, etc., including failed attempts. Baseball players currently in a match proactively prepare to hit or catch the ball with clearer action patterns (lower deviation) compared with the former situation. Furthermore, delayed motor intention (thinking about physical action carried out in the future) was found to be represented through a goal-based mechanism, allowing the mind to flexibly generate adapted action based on environmental constraints [96].

It is important to emphasize the fact that simulated feedback cannot replace objective feedback in terms of adaptation's base effectiveness. A system using a proactive optimization approach that only relies on simulated feedback while disregarding objective feedback is less effective than a system using both types. Additionally, as mentioned, optimization relying on simulation is prone to deviation due to self-referencing and self-reinforcing subjective values. For example, people can develop hallucinations if they are subjected to extended visual sensory deprivation such as blindfolding [97]. Simulated interactions and feedback can speed up the process of fitting to a deviated simulated objective sphere, which is consistent with findings showing that higher rumination may drive higher delusion proneness [98], predicting delusional and hallucinatory experience [99], as well as amplifying the association between negative effects and paranoia [100]. The same mechanism can also be seen in some harmful tendencies on collective levels. For example, in-group ideological reinforcement and a lack of information exchange with out-groups are strong drivers for radicalization and discrimination [101].

As a system absorb more information from the external environment, its entropy increases. A biological system needs to spend energy to maintain its highly ordered structure. On a collective level, the Earth's biosphere utilizes energy from the Sun to decrease its internal entropy as life forms continue to exchange information, grow, and evolve into more complex systems [49]. With the goal of maximizing energy usage effectiveness, the subjective sphere is constantly optimized to fit reality as much as possible since deviation means extra energy for fixing errors. In other words, introducing new, unfamiliar values to a mind puts it into an initial state of natural stupidity regarding those values. In advanced systems, generating new, untested simulated values puts the mind into an initial state of natural delusion. Because any biological system keeps interacting with the changing external environment, the optimization for adaptation never stops as long as the system is functional [27]. In humans, a certain degree of subjective sphere deviation allows for mental mutations (novel combinations) to happen in mental simulation, which is the prerequisite for creativity. However, when deviation becomes too high (stupid or delusional), the mind becomes ineffective at its information-processing functions. The energy trade-off implies that "bigger and better" innovations require more energy to fit the subjective sphere along the process. This is an underlying level in the philosophy of the cost of science [102] that the public may overlook (how much resources equal a valuable idea).

Due to neuroplasticity, the updating processes in human minds are "live-wiring" as opposed to the dominant "hard-wiring" manner in simpler systems (e.g., more dependent on predetermined genetic information) [25]. For humans, information absorbed from the environment and integrated into the mindset is stored in the form of trusted values (beliefs) [10]. To adapt to the changing external environment, beliefs are dynamically updated from related experiences, including newly received information and generated thoughts. In other words, the content of the system shifts in order to fit the subjective sphere to the objective sphere on a continuous timeline. Bayes' Theorem (presented below) is helpful when examining the information process of belief updating.

$$p(A|B) = \frac{p(B|A)p(A)}{p(B)}$$

This can be interpreted as follows: the posterior probability distribution is proportional to the prior probability distribution and the likelihood function. Collective evolution and individual learning can both be considered optimization processes, in which equivalence between certain evolutionary dynamics and Bayesian inference helps shed light on the progress of human cognition [103]. In fact, due to the similarity in information processing between Bayesian inference and human cognition (e.g., dealing with uncertainty due to information incompleteness, updating manner, etc.), Bayesian modeling is a useful tool in examining the natural working of the human mind [104,105]. This is also one of the main reasons why Bayesian statistics and the mindsponge theory have high compatibility, as shown in the research method of Bayesian Mindsponge Framework (BMF) analytics [106,107].

## 3. Empirical Validation

### 3.1. Study Overview

In this section, we provide empirical evidence to validate the effectiveness of the ecomindsponge conceptual framework in studying human psychology and behavior in relation to nature. Specifically, we examined how four different subjective spheres influence the attitudes toward biodiversity loss prevention measures among urban residents in Vietnam. The examination was conducted employing the Bayesian Mindsponge Framework (BMF) analytics on a dataset of 535 urban residents across Vietnam.

The interactions between urban residents and biodiversity loss are examined for several reasons. Biosphere integrity is one of two core planetary boundaries that help define a "safe operating space" for human societies development but is currently degrading at an unprecedented rate [108]. Urban residents are not only the drivers of biodiversity loss but also the actors preventing biodiversity decline and supporting conservation efforts. On the one hand, the unsustainable consumption demand of wildlife products in urban areas (e.g., food and medicinal purposes) is a major threat to the richness and endemism of species [109–113]. On the other hand, urban residents have both the desire and capacity to finance conservation efforts through nature-based tourism if appropriately managed [4,114,115]. Therefore, the interaction between urban residents and biodiversity loss is a typical relationship between humans and nature.

Why is the relationship of biodiversity loss with urban people used to validate the ecomindsponge conceptual framework, but not that with rural people? It is because of the contextual factors of cities. Urban residents are composed of people with various backgrounds, experiences, and pools of knowledge in relation to nature, increasing the existence likelihood of the subjective spheres of influence's special scenario: people with no spheres of influence and being influenced by the biodiversity loss. People falling into that special scenario are often urban residents having little knowledge and experience with nature. For example, Vietnamese young people who were born in the cities might likely belong to that special scenario, as they have few chances to interact with nature and limited knowledge about biodiversity due to the absence of nature education at school and within the family.

### 3.2. Materials and Methods

The dataset analyzed in the current study consists of responses from 535 Vietnamese urban residents about their multifaceted perceptions and interactions with biodiversity-related concepts [116]. The dataset is deposited in the Science Data Bank repository (https://doi.org/10.11922/sciencedb.j00104.00097), while its data descriptor is available at MIT Data Intelligence (https://doi.org/10.1162/dint_a_00110). The responses were conducted through a web-based survey via Google Forms from 18 June to 8 August 2021. The survey was designed based on the interview results with 38 urban residents in the two largest cities in Vietnam (Ho Chi Minh and Hanoi) as well as the mindsponge theory and ecomindsponge conceptual framework.

This study was performed in line with the principles of the Declaration of Helsinki. Before filling out the questionnaire, respondents were asked to read and agree with the consent form, which specifies the study aims, questionnaire information, and participant

confidentiality. Because this study was not funded, we are not bound by any contractual duties and may completely prioritize the obligation to protect participants. Ethical approval is not required by our institutes for social survey research. Moreover, to the best of our knowledge, formal ethical review boards of ethics in conservation research practice are unavailable in Vietnam due to the lack of expertise and resources.

For performing the analysis, seven variables were retrieved from the original dataset (see Table 2). Five variables representing respondents' agreement on whether species conservation (*B4_1*), environmental law (*B4_3*), environmental tax (*B4_8*), donation for biodiversity conservation (*B4_9*), and prohibition of illegal wildlife consumption (*B4_7*) were used as outcome variables. Meanwhile, the remaining two variables (*B5_1* and *B6_1*) were employed to generate a categorical variable demonstrating four fundamental scenarios of the respondents' subjective spheres of influence in relation to biodiversity loss.

**Table 2.** Variable description.

| Variable | Original Variable | Meaning | Type of Variable | Value |
|---|---|---|---|---|
| *Conservation* | *B4_1* | Whether the respondent supports species conservation as a preventive measure against biodiversity loss | Numerical | Ranging from 1 (strongly disagree) to 4 (strongly agree) |
| *EnvironmentalLaw* | *B4_3* | Whether the respondent supports environmental law enactment as a preventive measure against biodiversity loss | Numerical | Ranging from 1 (strongly disagree) to 4 (strongly agree) |
| EnvironmentalTax | *B4_8* | Whether the respondent supports environmental tax as a preventive measure against biodiversity loss | Numerical | Ranging from 1 (strongly disagree) to 4 (strongly agree) |
| Donation | *B4_9* | Whether the respondent supports donation for biodiversity conservation as a preventive measure against biodiversity loss | Numerical | Ranging from 1 (strongly disagree) to 4 (strongly agree) |
| WildConsProhi | *B4_7* | Whether the respondent supports the prohibition of illegal wildlife consumption as a preventive measure against biodiversity loss | Numerical | Ranging from 1 (strongly disagree) to 4 (strongly agree) |
| *SphereofInfluence* | N/A | Generated from variables *B5_1* and *B6_1* | Categorical | Both spheres = 1 Sphere of being influenced = 2 Sphere of influence = 3 No sphere = 4 |
| N/A | *B5_1* | Agreement with that the following object is affected by biodiversity loss (My life) | Binary | Agree = 1 Disagree = 0 |
| N/A | *B6_1* | Agreement with that the following subject can contribute to biodiversity loss prevention (Myself) | Binary | Agree = 1 Disagree = 0 |

The categorization of spheres is presented in Table 3. To elaborate, variable *B5_1* indicates the perceived influence of biodiversity loss on the respondent, whereas variable *B6_1* indicates the perceived influence of the respondent to prevent biodiversity loss. Both variables are measured by a 4-point Likert scale, ranging from "strongly disagree" to "strongly agree". We modified the two variables to binary variables to aid the categorization of spheres by grouping "strongly disagree" and "disagree" as "disagree" and "strongly agree" and "agree" as "agree". Then, a new categorical variable with four levels was created from these two binary variables: "both spheres", "sphere of being influenced", "sphere of influence", and "no sphere".

Bayesian multilevel modeling was employed to analyze the data for several reasons. First, the ecomindsponge conceptual framework, specifically the subjective spheres of influence, was developed from the mindsponge theory, so it also follows the set theory logic. The adoption of set theory logic makes ecomindsponge-based models compatible with hierarchical model construction [106,107]. Moreover, we attempted to validate the effectiveness of subjective spheres of influence in predicting people's attitudes towards

biodiversity loss prevention methods, so parsimonious models are prioritized. Bayesian analysis treats all properties (including unknown variables) probabilistically [117], so it helps fit the models while ensuring the parsimony principle, making Bayesian multilevel modeling suitable with our aim.

**Table 3.** Categorization of urban residents' subjective spheres of influence.

| | | **B5_1: Agreement with That the Following Object Is Affected by Biodiversity Loss (My Life)** | |
| | | **"Agree"** | **"Disagree"** |
| --- | --- | --- | --- |
| *B6_1*: Agreement with that the following subject can contribute to biodiversity loss prevention (Myself) | "Agree" | A person with both spheres | A person with only the sphere of influence |
| | "Disagree" | A person with only the sphere of being influenced | A person with no sphere at all |

Besides the compatibility with the ecomindsponge conceptual framework, Bayesian multilevel modeling was also employed because of its advantages in treating categorical data. The categorical variable *SphereofInfluence* has four clusters, but there exists an imbalance between the number of observations across clusters: (1) "both spheres" level accounts for 83.37% of observations; (2) "sphere of being influenced" level accounts for 0.75%; (3) "sphere of influence" level accounts for 11.59%; and (4) "no sphere" accounts for 4.30%. For such substantial imbalances across levels, multilevel modeling can help prevent over-sampled clusters from dominating the inference [118]. In addition to that, multilevel modeling also supports estimating of the variation among clusters explicitly, which is suitable for studying the impacts of each subjective sphere scenario on urban residents' attitudes [119].

We employed the varying intercepts model to estimate the effect of each subjective sphere scenario on urban residents' attitudes. Varying intercepts assign a unique intercept parameter for each cluster of the data (or each subjective sphere scenario). In the following models, *SphereofInfluence* is used as a predictor variable, while *Conservation*, *EnvironmentalLaw*, *EnvironmentalTax*, *Donation*, and *WildConsProhi* were used as outcome variables, respectively:

$$Conservation \sim \alpha[SphereofInfluence] \tag{1}$$

$$EnvironmentalLaw \sim \alpha[SphereofInfluence] \tag{2}$$

$$EnvironmentalTax \sim \alpha[SphereofInfluence] \tag{3}$$

$$Donation \sim \alpha[SphereofInfluence] \tag{4}$$

$$WildConsProhi \sim \alpha[SphereofInfluence] \tag{5}$$

Based on the reasoning above of subjective spheres of influence, we expected respondents with "both spheres" to have the highest scores of agreements towards biodiversity loss prevention measures, while respondents with "no sphere" to have the lowest scores (as measured by intercept values). It was also expected that people with only one sphere have lower scores than people with "both spheres" and higher scores than people with "no sphere". Furthermore, the intercept values of people with a "sphere of being influenced" have greater variance than those with a "sphere of influence". If these expectations are confirmed, the effectiveness of subjective spheres of influence in affecting human psychology in relation to nature.

All the Bayesian multilevel analyses were conducted using the bayesvl R package [120]. The package offers researchers a user-friendly and intuitive protocol, the ability to visualize beautiful graphics, and cost-effectiveness [121,122]. Model fitting was conducted with four Markov chains with 6000 iterations for each chain. The first 2000 iterations were installed as a warmup period. As the current study is exploratory research, uninformative priors were employed to avoid subjective biases.

### 3.3. Results

The estimated results of all models are displayed in this subsection. Before interpreting the posteriors, it is necessary to check for the Markov chains convergence of each model. Two fundamental statistics used to evaluate the convergence are the effective sample size (*n_eff*) and (*Rhat*). A model's Markov chains are deemed convergent when the parameters' *n_eff* values are larger than 1000 [118], and *Rhat* values are equal to 1 [123]. From Table 4, it can be seen that all *n_eff* values are greater than 1000, and *Rhat* values are equal to 1, so Markov chains can be considered to be convergent.

**Table 4.** Estimated posterior results.

| Model 1: *Conservation ~ α[SphereofInfluence]* | | | | |
|---|---|---|---|---|
| **Parameters** | **Mean** | **SD** | **n_eff** | **Rhat** |
| a_SphereofInfluence[BothSphere] | 3.46 | 0.03 | 16,421 | 1 |
| a_SphereofInfluence[SphereBeingInfluenced] | 2.76 | 0.31 | 15,156 | 1 |
| a_SphereofInfluence[SphereInfluence] | 3.18 | 0.08 | 17,324 | 1 |
| a_SphereofInfluence[NoSphere] | 1.89 | 0.13 | 16,157 | 1 |
| a0_SphereofInfluence | 2.83 | 0.86 | 1254 | 1 |
| sigma_SphereofInfluence | 1.34 | 1.10 | 1754 | 1 |
| **Model 2:** *EnvironmentalLaw ~ α[SphereofInfluence]* | | | | |
| **Parameters** | **Mean** | **SD** | **n_eff** | **Rhat** |
| a_SphereofInfluence[BothSphere] | 3.62 | 0.03 | 15,552 | 1 |
| a_SphereofInfluence[SphereBeingInfluenced] | 3.24 | 0.28 | 15,998 | 1 |
| a_SphereofInfluence[SphereInfluence] | 3.55 | 0.07 | 17,214 | 1 |
| a_SphereofInfluence[NoSphere] | 2.11 | 0.12 | 15,637 | 1 |
| a0_SphereofInfluence | 3.09 | 0.85 | 1237 | 1 |
| sigma_SphereofInfluence | 1.36 | 1.16 | 2135 | 1 |
| **Model 3:** *EnvironmentalTax ~ α[SphereofInfluence]* | | | | |
| **Parameters** | **Mean** | **SD** | **n_eff** | **Rhat** |
| a_SphereofInfluence[BothSphere] | 3.33 | 0.03 | 16,851 | 1 |
| a_SphereofInfluence[SphereBeingInfluenced] | 2.76 | 0.33 | 16,711 | 1 |
| a_SphereofInfluence[SphereInfluence] | 3.09 | 0.09 | 16,975 | 1 |
| a_SphereofInfluence[NoSphere] | 2.04 | 0.15 | 16,247 | 1 |
| a0_SphereofInfluence | 2.76 | 0.89 | 1003 | 1 |
| sigma_SphereofInfluence | 1.20 | 1.17 | 1510 | 1 |
| **Model 4:** *Donation ~ α[SphereofInfluence]* | | | | |
| **Parameters** | **Mean** | **SD** | **n_eff** | **Rhat** |
| a_SphereofInfluence[BothSphere] | 3.35 | 0.03 | 15,841 | 1 |
| a_SphereofInfluence[SphereBeingInfluenced] | 2.76 | 0.31 | 15,636 | 1 |
| a_SphereofInfluence[SphereInfluence] | 3.00 | 0.09 | 15,875 | 1 |
| a_SphereofInfluence[NoSphere] | 2.16 | 0.15 | 15,204 | 1 |
| a0_SphereofInfluence | 2.80 | 0.62 | 1666 | 1 |
| sigma_SphereofInfluence | 1.02 | 0.90 | 2384 | 1 |
| **Model 5:** *WildConsProhi ~ α[SphereofInfluence]* | | | | |
| **Parameters** | **Mean** | **SD** | **n_eff** | **Rhat** |
| a_SphereofInfluence[BothSphere] | 3.35 | 0.03 | 15,485 | 1 |
| a_SphereofInfluence[SphereBeingInfluenced] | 2.77 | 0.32 | 15,134 | 1 |
| a_SphereofInfluence[SphereInfluence] | 3.00 | 0.09 | 15,574 | 1 |
| a_SphereofInfluence[NoSphere] | 2.16 | 0.15 | 15,187 | 1 |
| a0_SphereofInfluence | 2.81 | 0.61 | 1037 | 1 |
| sigma_SphereofInfluence | 1.02 | 0.90 | 2340 | 1 |

We also diagnosed the convergence, or the Markov property, of all models using trace plots, Gelman–Rubin–Brook plots, and autocorrelation plots. The trace plots of Models 1

to 5 show "healthy" stochastic simulation processes, which are stationary and centralized around an equilibrium (see Appendix A, Figures A1, A4, A7, A10 and A13). The shrink factors in the Gelman–Rubin–Brook plots of Models 1 to 5 drop swiftly to 1 during the warmup period (see Appendix A, Figures A2, A5, A8, A11 and A14), while the parameters' autocorrelation levels in the autocorrelation plots decline rapidly to 0 after a finite number of lags (see Appendix A, Figures A3, A6, A9, A12 and A15). These demonstrations signify the good convergence of Markov chains.

Based on the estimated results of Models 1–5, we discovered that people with both spheres have the greatest attitudes towards biodiversity loss prevention measures, and people with no spheres have the poorest attitudes. Meanwhile, people with only one of either sphere were found to have lower scores than people with both spheres and higher scores than people with no spheres. The probability distributions of the intercepts in Figure 4 confirm these patterns. Moreover, Figure 4 also shows a much higher variance of $\alpha_{SphereofInfluence[SphereBeingInfluenced]}$'s distribution than that of $\alpha_{SphereofInfluence[SphereInfluence]}$'s distribution. All these results support the validity of the ecomindsponge conceptual framework's assumptions about subjective spheres of influence.

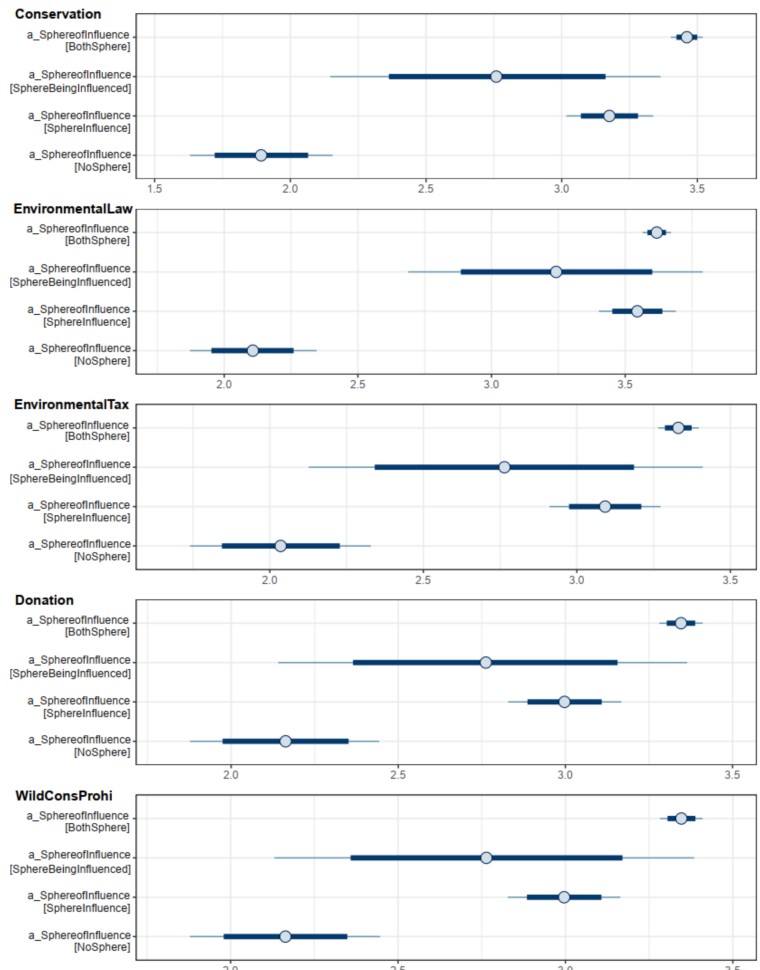

**Figure 4.** Posterior distributions of the models.

*3.4. Discussion*

Our statistical analyses show consistent patterns of how the subjective spheres of influence and being influenced affect people's responses toward environmental goals. Those who perceive both sides of the interaction have the strongest and clearest positive responses. Those who perceive no impacts from either side have the weakest responses. Among the one-sided perceptions, those who only think they are impacted by biodiversity loss have weaker responses with greater variance compared with those who only think they

can impact the state of biodiversity loss. The evidence supports our reasoning about how different mindset types affect decision-making based on the presented subjective sphere optimization mechanisms.

Following the mindsponge mechanism of information processing, values are accepted (and then expressed as attitudes or behaviors) only when they have a net positive perceived benefit. In other words, after (subjectively) considering all currently available negative and positive meanings attached to a value, the final sum is positive. In the nature-human relationship, both sides of influence can have negative and positive meanings. For example, being influenced by nature can include obtaining benefits from natural resources and various ecological services as well as suffering from natural disasters and diseases. Influencing nature can include beneficial actions such as healing damaged ecosystems and conserving endangered species as well as destructive actions such as deforestation, producing pollutants, and causing extinctions. Here, it is crucial to note that the same action can mean positive for someone and negative for another. For example, the act of hunting wild elephants is considered negative for rangers and activists but positive for poachers and illegal consumers. The non-straightforward psychological processes related to consuming bushmeat and environmental attitude were examined in a former study [109]. People's attitudes and intentions toward avoiding air pollution also involve considerations of different psychosocial factors [124]. The meanings attached to information are heavily dependent on cultural values, self-interests, and personal beliefs [10].

Attitudes (as sets of beliefs and intentions) can induce corresponding behaviors when the intensity of the values passes certain thresholds. The binary switches in decisions for carrying out actions are determined by the gradient accumulation of belief strength. Even extreme behaviors (such as suicide and homicide) follow this mindsponge mechanism [101]. Thus, when considering how subjective spheres affect human behaviors, it is important to examine the sociocultural characteristics, for society is a major part of the infosphere surrounding modern humans. Information about the objective world (such as climate change and biodiversity loss) mainly comes from intermediate transmitters (e.g., interpersonal communication, the Internet, books, and other media forms, etc.).

Again, it should be emphasized that besides objective availability of information, subjective accessibility and effective rate of absorption need to be paid attention to in environmental communication efforts. Trust in information sources is especially important due to its "gatekeeper" function in information processes [55,84]. Available information sources may not be accessed if they are associated with negative meanings, such as inconvenience or stigma [125]. Strong negative preconceptions about a specific value can make a person hastily disregard all incoming information carrying opposite values, regardless of information availability and accessibility. This aspect is clear in the cases of misinformation and conspiracy theories during the COVID-19 pandemic [126,127].

Building an eco-surplus culture is creating a more pro-environmental infosphere [3,4,128–130]. In such an infosphere, pro-environmental values are reinforced based on exposure, transmission frequency, and the mechanism of mindset updating, where the filtering system use newly integrated values as references for subsequent related evaluation. Moreover, following the mindsponge mechanism of trust, pro-environmental social norms form in an eco-surplus culture, serving as collective trust and making it harder for misinformation and disinformation to affect the collective mindset [10]. This is the same natural mechanism the mind uses to protect itself from shock due to exposure to unfamiliar values [27]. Additionally, due to the self-regulation quality of society as an information system, a collective eco-friendly mindset finds it easier to adjust other aspects accordingly, such as finance, laws, entertainment, aesthetics, ethics, etc., without much cognitive dissonance due to conflicts with self-interests. The endeavors of building an eco-surplus culture can be performed by using the functions of subjective cost-benefit judgments (including rationality and emotion) besides providing information accessibility. For example, people should understand the consequences of biodiversity loss correctly to establish the attached costs [4]. Feedback of beliefs should come from real events and statis-

tics to avoid subjective sphere deviation. In summary, we fight stupidity with information availability and accessibility and fight delusion with information interpretation capacity and processing feedback based on reality.

Considering the importance of transparency toward scientific progress and the issue of cost in science [102,131], we disclose the limitations of our evidence as follows. The evidence provided in the present study only shows the applicability of subjective spheres in thinking (intentions and attitudes). Further studies should focus on applying subjective spheres to predicting or finding patterns in behaviors. Reaching the threshold for carrying out actions may require more complex interactions among factors within the subjective spheres in relation to both society and nature. For example, behaviors regardless of cognitive dissonance due to conflicting values under high buffering capacity are a typical example of the cultural additivity phenomenon [132]. Additionally, our study only used a sample of Vietnamese urban residents. Further studies using samples from other countries, regions, or special populations can be used to update and adjust our results. The Bayesian approach for statistical analysis is advantageous in this regard due to its fundamental updating manner using priors.

## 4. Conclusions and Further Development Directions for Ecomindsponge

The proposed ecomindsponge conceptual framework offers a novel way of thinking about the position of humans in the ecosphere. The nature-human relationship was examined from the metaphysical subject-object perspective using logic methods and new evidence from natural sciences. On this basis, we derived and constructed a conceptual framework of system boundaries, selective exchange, and adaptive optimization based on information processing principles. Testing the conceptual properties on real data showed reliable consistency. This also demonstrates the framework's applicability to real-world problems, focusing on but not limited to environmental issues.

Additionally, in this section, we share some further thoughts on some promising applications, especially through the study of human culture and innovation. We also briefly present opinions on the direction of interdisciplinary social research with the information processing approach.

For humans, culture and innovation are two crucial aspects of survival and adaptation [133,134]. Culture helps maintain social order and regulate interpersonal relationships, which enables the development of other delicate human constructs such as morality, academics, the arts, philosophy, etc. in comparatively safe and stable environments. Innovation improves humans' information processing power and helps increase the effectiveness of adaptation [27,135]. If applied wisely, the objective sphere of influence can be expanded through effective information exchange, and subjective sphere optimization is also improved through better simulation capability.

Culture can be considered a collective set of trusted values expressed in the form of social norms. It represents all information within the collective subjective sphere of influence that a society bases upon to evaluate any value existing within and coming into the system. Human culture, thus, determines how the majority of humans think of and conduct behaviors toward nature. The direction of collective evaluation is reflected as an eco-surplus or eco-deficit culture [3]. Collective values keep changing to adapt, regardless of strategies and whether the outcome is adaptive or maladaptive (e.g., environmentally friendly or destructive). This property can also be seen in psychosocial phenomena such as cultural additivity [132], transnational identity shifting [127], and lie-violence value integration [136].

Humans utilize knowledge to proactively induce changes that are expected to be beneficial, which is the nature of innovation. The base knowledge, however, all comes from nature, the system of which humanity is a subset. The aspects of set boundary and information incompleteness are presented in the sections above. Suppose the contents and patterns of thoughts have a high level of deviation and do not follow natural principles. In that case, they are likely not useful when applied to the real world through intentional behaviors.

Expanding the subjective sphere of influence means having awareness of a wider range of information exchange. Awareness of more "links" (interactions) can help one gather priorly unknown knowledge from the external world, increasing the accuracy and efficiency of subjective sphere optimization. In brief, by becoming more capable and more aware of that capability, humanity can learn more from nature, the ultimate intelligence that made us what we are [128,137].

**Author Contributions:** Conceptualization, M.-H.N. and T.-T.L.; methodology, M.-H.N.; software, Q.-H.V.; validation, Q.-H.V.; formal analysis, M.-H.N.; investigation, T.-T.L.; resources, M.-H.N.; data curation, M.-H.N.; writing—original draft preparation, M.-H.N. and T.-T.L.; writing—review and editing, T.-T.L.; visualization, M.-H.N. and T.-T.L.; supervision, Q.-H.V.; project administration, M.-H.N. All authors have read and agreed to the published version of the manuscript.

**Funding:** This research received no external funding.

**Institutional Review Board Statement:** Not applicable.

**Informed Consent Statement:** Not applicable.

**Data Availability Statement:** The dataset is deposited in the Science Data Bank repository (https://doi.org/10.11922/sciencedb.j00104.00097), while its data descriptor is available at MIT Data Intelligence (https://doi.org/10.1162/dint_a_00110).

**Conflicts of Interest:** The authors declare no conflict of interest.

**Ethical Statement:** As this study did not receive funding, we are not bounded by any contractual responsibilities and can fully prioritize the obligations to protect participants. Ethical approval is not required by our institutes for social survey research. Moreover, to the best of our knowledge, formal ethical review boards of ethics in conservation research practice are unavailable in Vietnam due to the lack of expertise and resources. Informed consent was obtained from all participants. Before filling out the questionnaire, respondents were asked to read and agree with the consent form, which specifies the study aims, questionnaire information, and participant confidentiality.

## Appendix A

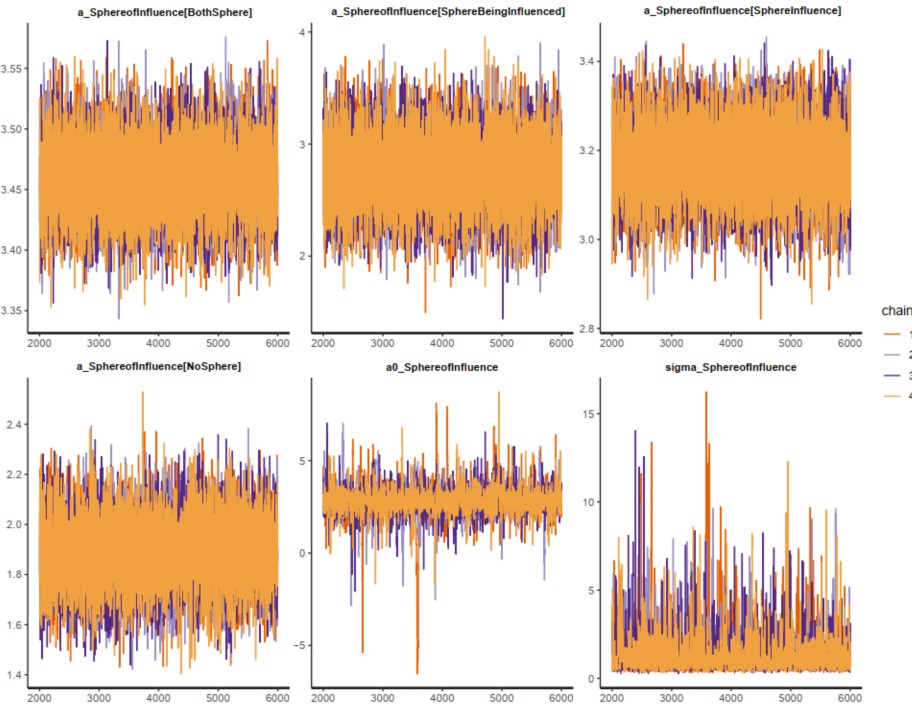

**Figure A1.** Model 1's trace plots.

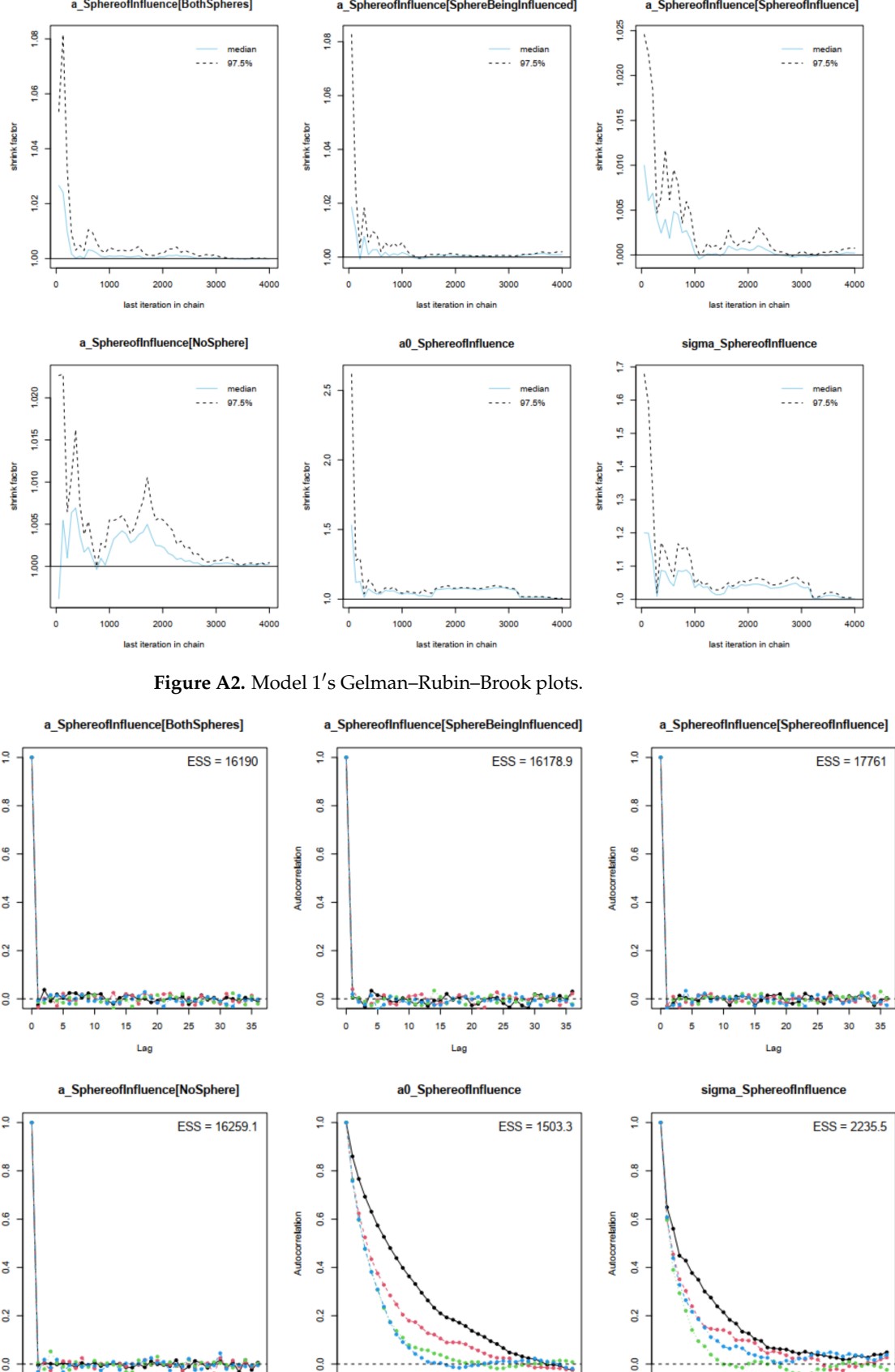

**Figure A2.** Model 1′s Gelman–Rubin–Brook plots.

**Figure A3.** Model 1′s autocorrelation plots.

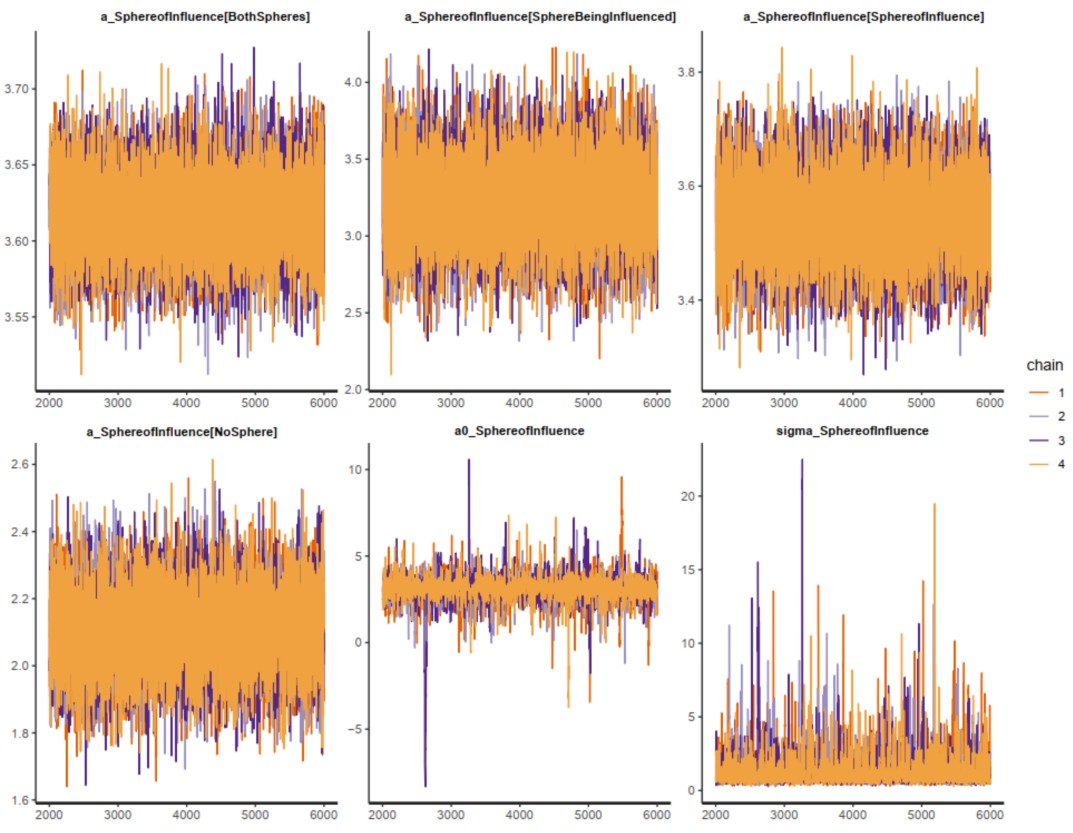

**Figure A4.** Model 2′s trace plots.

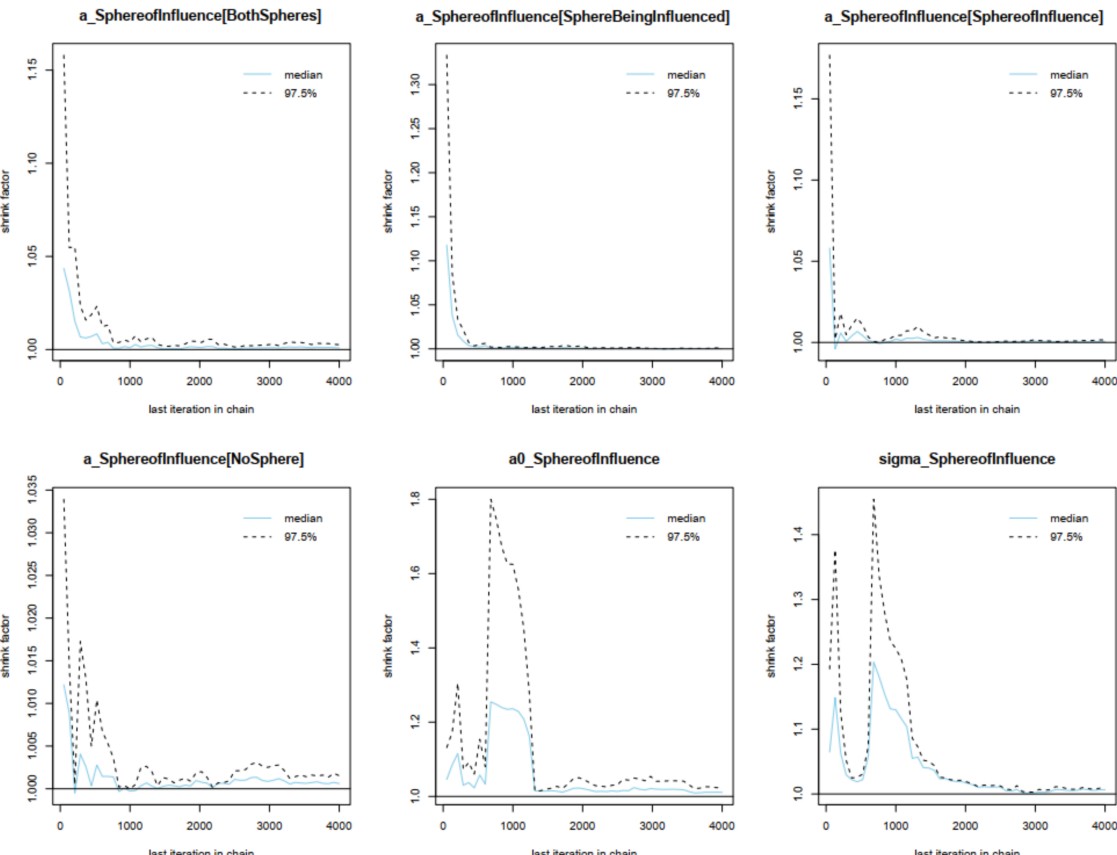

**Figure A5.** Model 2′ Gelman–Rubin–Brook plots.

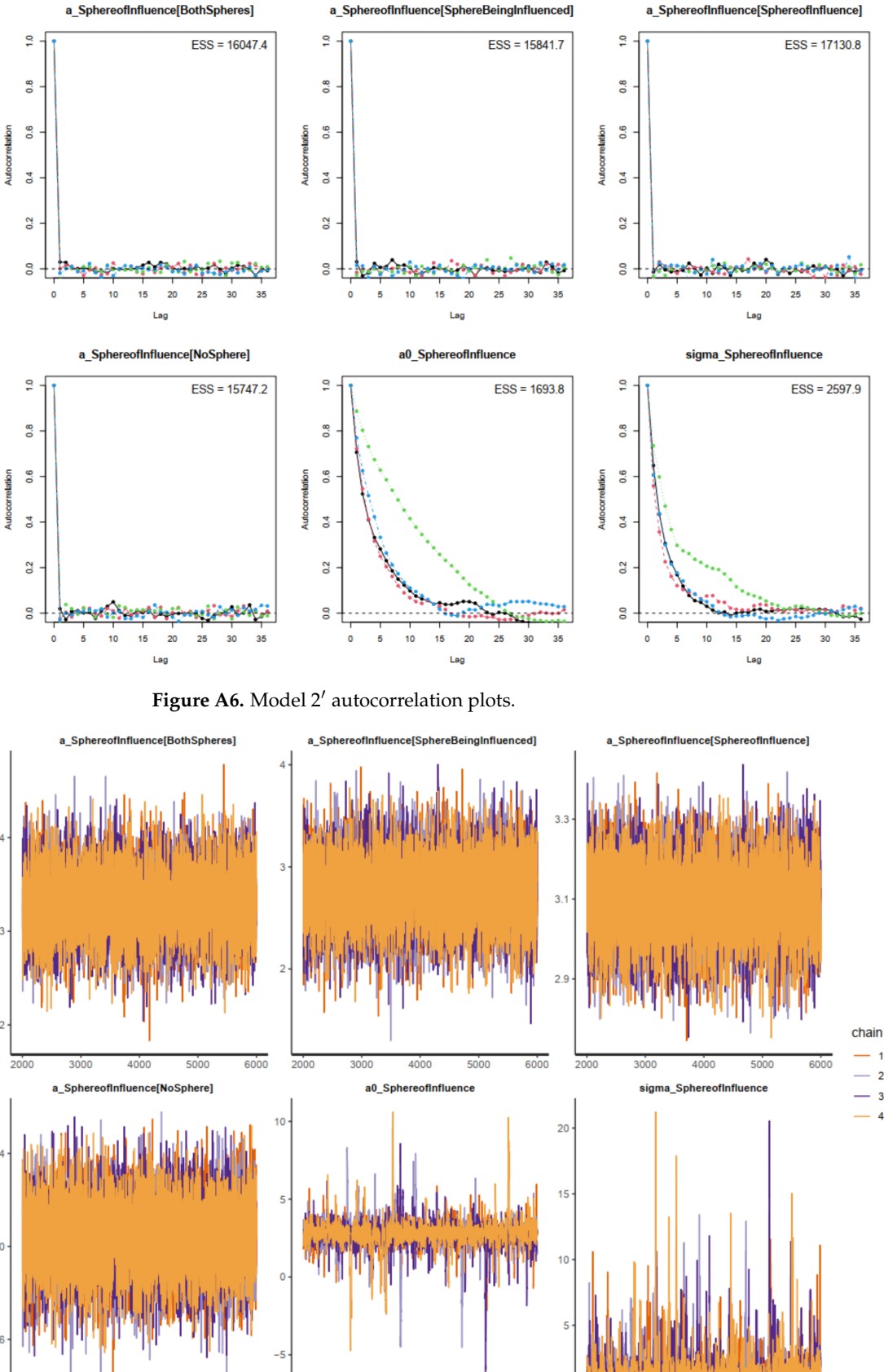

**Figure A6.** Model 2′ autocorrelation plots.

**Figure A7.** Model 3′s trace plots.

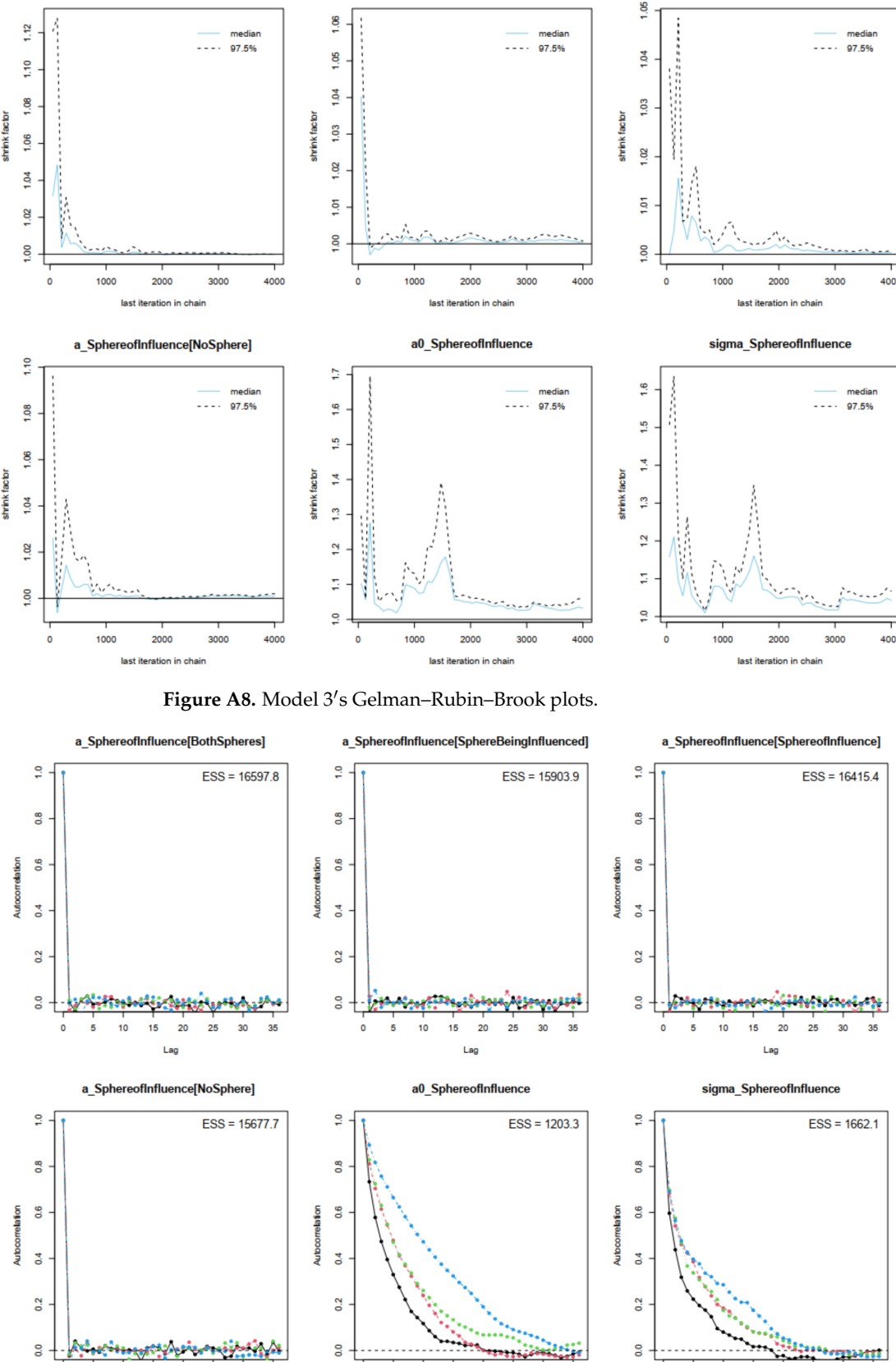

**Figure A8.** Model 3′s Gelman–Rubin–Brook plots.

**Figure A9.** Model 3′s autocorrelation plots.

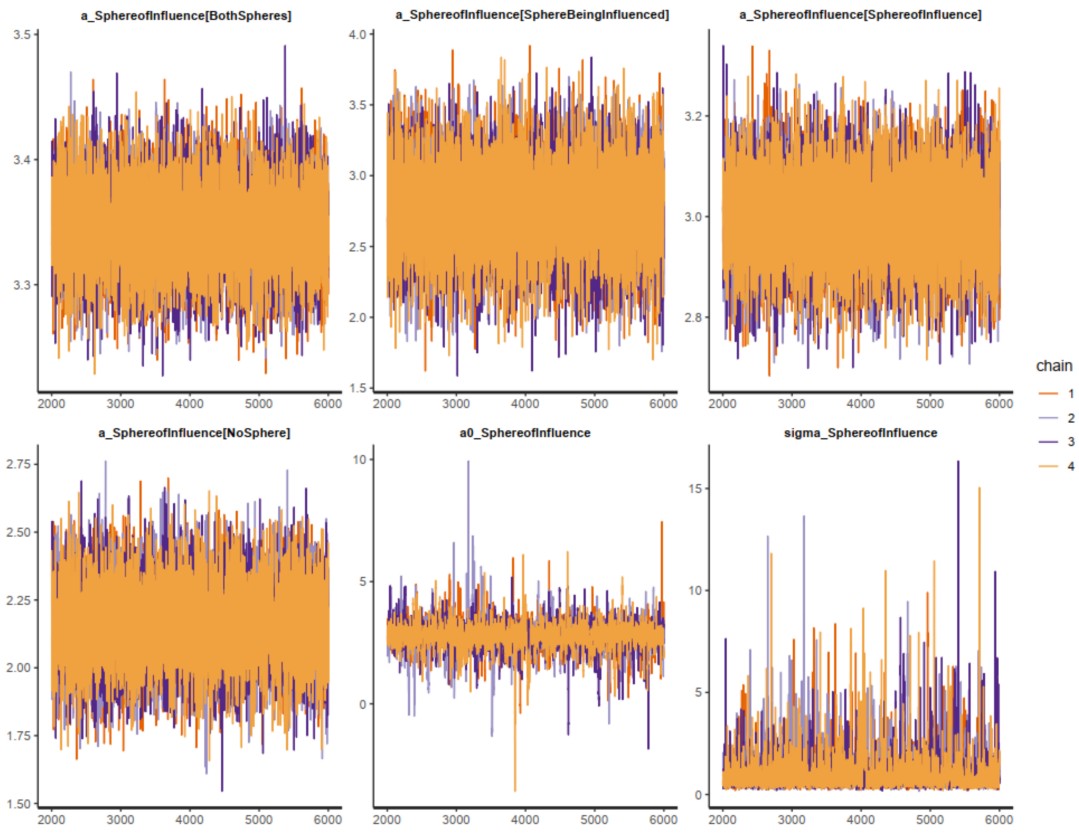

**Figure A10.** Model 4′s trace plots.

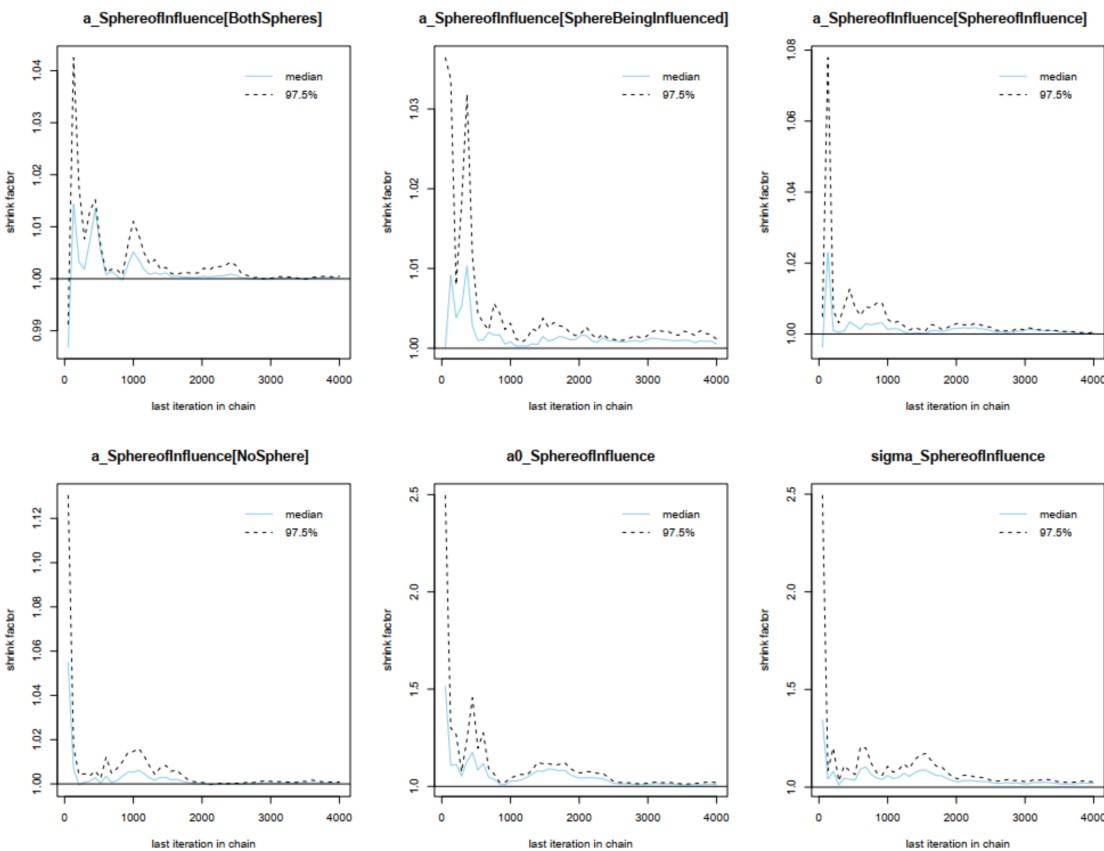

**Figure A11.** Model 4′s Gelman–Rubin–Brooks plots.

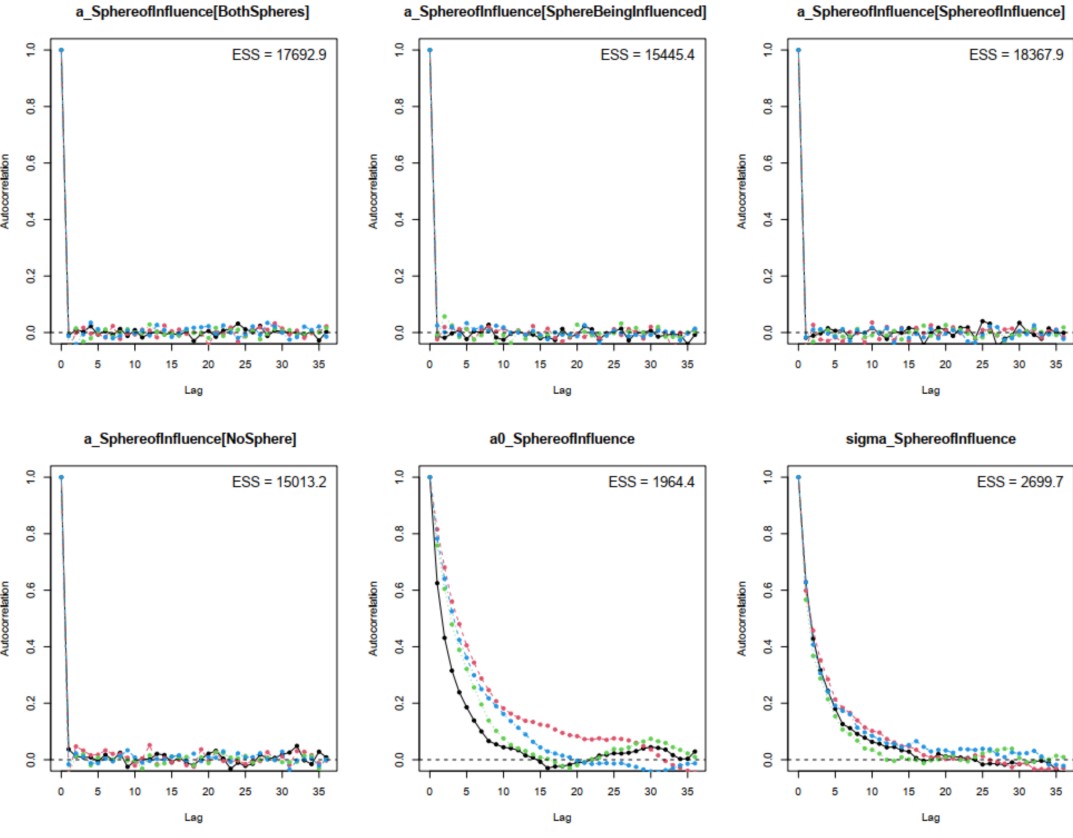

**Figure A12.** Model 4′s autocorrelation plots.

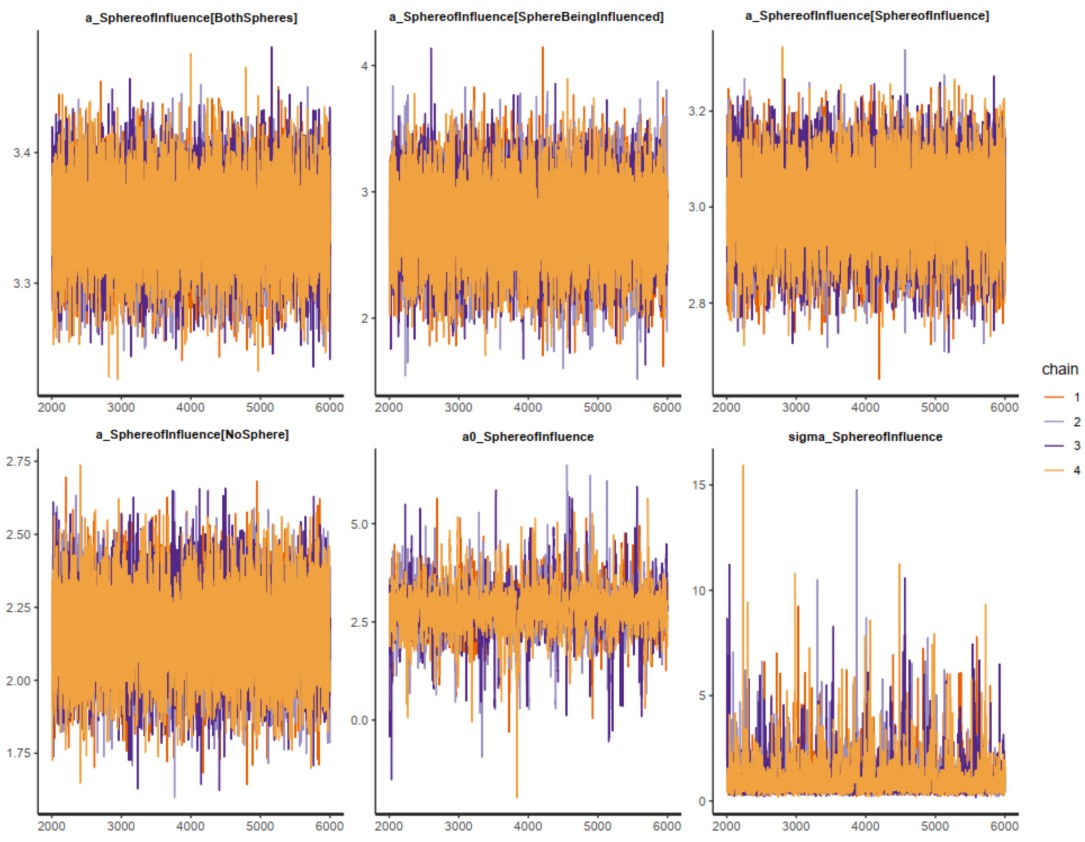

**Figure A13.** Model 5′s trace plots.

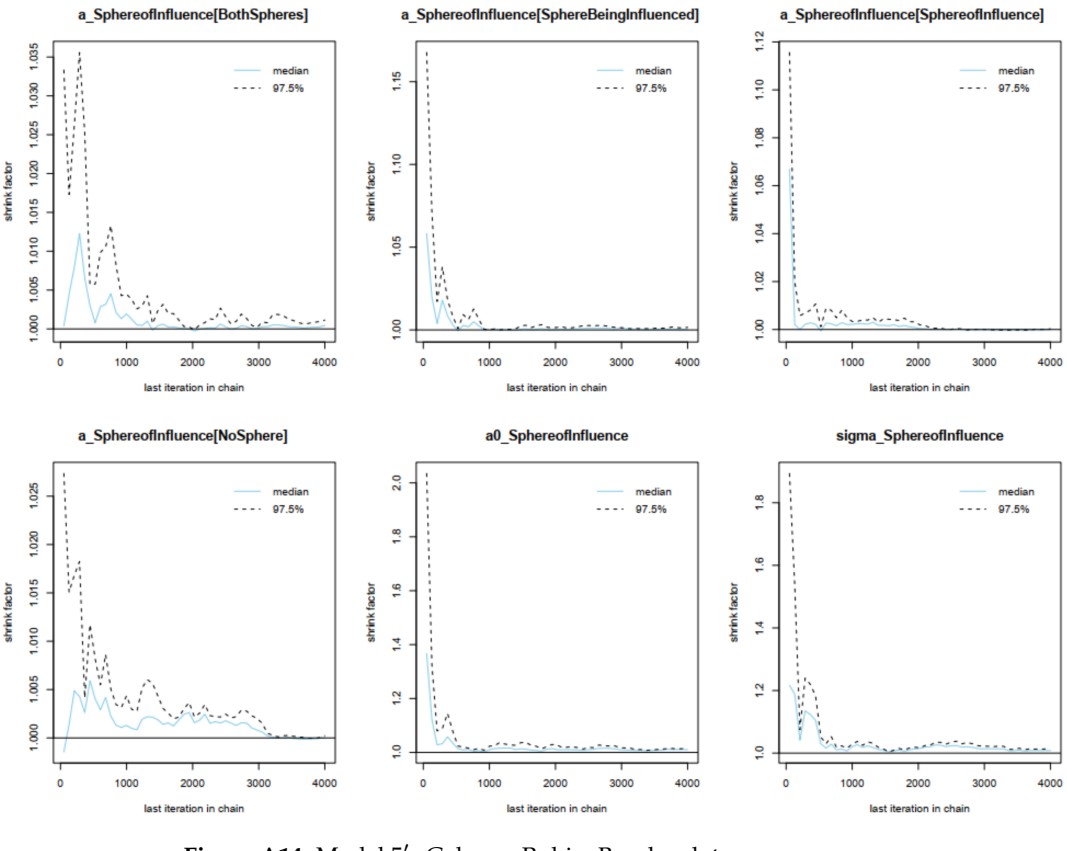

**Figure A14.** Model 5′s Gelman–Rubin–Brooks plots.

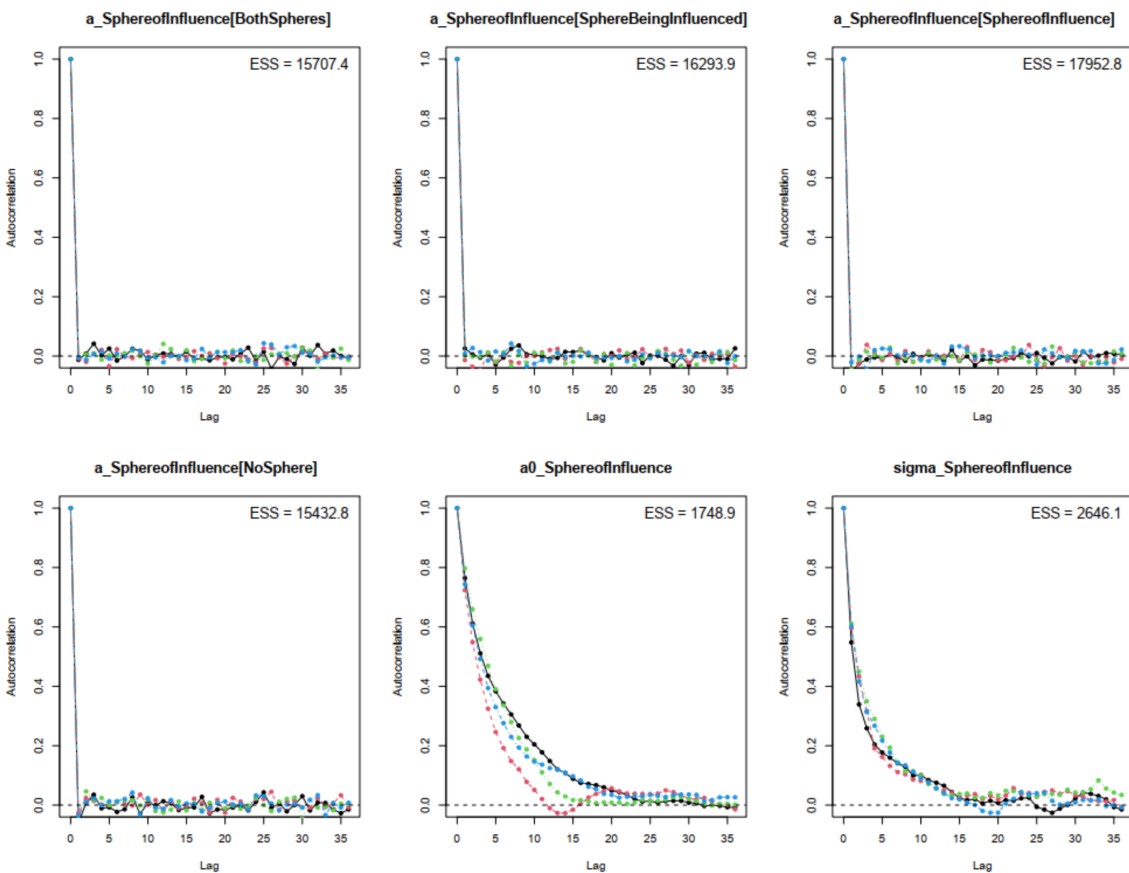

**Figure A15.** Model 5′s autocorrelation plots.

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
