# Peer review of "Ecomindsponge: A Novel Perspective on Human Psychology and Behavior in the Ecosystem"

_urbansci, doi:10.3390/urbansci7010031_

Round 1

Reviewer 1 Report

Good quality article, the topic is relevant not only for a broad academic community, but also for practitioners and other relevant stakeholders. The structure of the article is consistent with the requirements (the abstract disclose the contents of the main research results; the illustrations (tables, figures) are relevant). The article includes enough scientific analysis of literature and other sources on the subject of research. The research methods used are adequate and sufficient for this topic the results of the research reliable and well-founded. The conclusions in the article correct and logically grounded.

Reviewer 2 Report

An innovative approach which gives an alternative to the traditional psychometric models.The target to be objective,in a subjective approach,is fully accomplished.This work shows a facility to understand the human behaviour,combined with various sceneries of human mental approach.

Reviewer 3 Report

The author should revise their paper to answer the four questions below:

First question: What is the main challenge and issues in this study?

Second question: 'What is the criticism and gap analysis for academic literature that attempts to provide a solution?'

Third question: 'What is the recommended solution for such challenges and their issues?'

Fourth question: 'What are the implication, contributions, and novelty of the present study?'

Reviewer 4 Report

The author applied the widely a novel perspective on human psychology and behavior in the ecosystem. Having read this MS carefully, I have the minor concerns:

1. Some important information should be mentioned in the Introduction. For example, why you carried out this study? What aims you have by this research? What can its results be used for? Conservation implications, population management?
2. Why you focused on the variables? Need to give an introduction about the subject.
3. Analysis process need more detailed information.

4. It is not clear how the accuracy indexes of the analysis were calculated. Explain them clearly.
5. I suggest that, in the Discussion or Conclusion, some population management implications based on the results should be made.
